# BioFuse: an embedding fusion framework for biomedical foundation models

**Mirza Nasir Hossain**⬥*, **David Harris-Birtill**⬥

School of Computer Science, University of St Andrews, St Andrews, Fife, United Kingdom

⬥ Senior author.
* mnh3@st-andrews.ac.uk

## Abstract

The biomedical field has witnessed a surge in pre-trained foundation models excelling in specific subdomains such as radiology and histopathology. While integrating these models promises a more comprehensive understanding of biomedical data, it poses challenges in model compatibility and feature fusion. We present BioFuse, a novel open-source framework designed to generate optimised biomedical embeddings. BioFuse utilises a pool of 9 state-of-the-art (SOTA) foundation models to create task-specific embeddings. It employs grid search to automatically identify the optimal combination of models, fusing their embeddings through vector concatenation. On the MedMNIST+ benchmark, using XGBoost as the downstream classifier, BioFuse outperforms several existing methods, achieving SOTA AUC in 5/12 datasets, while maintaining near-SOTA performance across most remaining datasets. Remarkably, our experiments reveal unexpected cross-modal capabilities, with histopathology and radiology models showing strong performance when applied to other imaging modalities. BioFuse features a high-level API for immediate deployment and an extensible architecture to incorporate future models and fusion techniques. We anticipate BioFuse will not only enhance the utility of foundation models in biomedicine but also open new avenues for uncovering cross-modal relationships in biomedical data.

## 1. Introduction

Artificial Intelligence (AI) is driving advances in biomedical research and clinical practice [1–3]. Foundation models – large, pre-trained models capable of performing a wide variety of tasks with minimal adaptation – have emerged as key tools in this domain. Trained on vast, diverse datasets [4], these models have demonstrated promising performance in various biomedical subdomains [5].

In medical imaging, foundation models are used to detect and classify abnormalities in radiographs [5], generate diagnostic reports [6], segment images to delineate organs and tumours [7], analyse histopathology slides for disease diagnosis [8,9], and integrate information across imaging modalities to enhance diagnostic accuracy [10]. In

provided the original author and source are
credited.

**Data availability statement:** 1. MedMNIST+:
https://zenodo.org/records/10519652 2.
BioFuse Embedding Cache for MedMNIST+
2D Datasets: https://doi.org/10.5281/zeno-
do.16732578 3. BioFuse Embedding Cache for
ImageNet-1K: https://doi.org/10.5281/zeno-
do.14930584 4. BioFuse GitHub repository:
https://github.com/mnhcorp/biofuse.

**Funding:** The author(s) received no specific
funding for this work.

**Competing interests:** The authors have
declared that no competing interests exist.

clinical natural language processing (NLP), they are employed for clinical text summa-
rization [11] and information extraction [12], while in genomics, they help decipher the
language of non-coding DNA [13] and allow accurate cell type annotation [14].

## 1.1 Motivation

Typically, researchers manually select foundation models tailored for specific tasks
[15]. These models increasingly serve as feature extractors, transforming input data
into dense numerical representations called embeddings [16]. These embeddings
encapsulate complex features of the input data in high-dimensional vectors, allowing
efficient computation and comparison.

However, foundation models are often trained on single modalities, with research
groups developing specialized architectures for specific domains [17–19]. While
this approach enables exceptional performance within designed scopes, these
models may have learned features relevant beyond their original domains. As
domain-specific models become increasingly refined [20], there is a growing need to
explore their broader applicability across biomedical tasks and modalities.

Combining embeddings from multiple models allows us to exploit modality-specific
encodings while gaining new perspectives through cross-modal integration. Various
approaches have demonstrated the value of leveraging multiple modalities – for
instance, ConVIRT [21] utilises contrastive learning between paired images and text
to enhance chest X-ray classification performance. Such cross-modal techniques
suggest that strategic integration of specialized models could generate more com-
prehensive representations of biomedical data and unlock insights inaccessible to
single-modality approaches.

**This leads us to ask two key questions:**

1. Does combining the embeddings of multiple foundation models trained on differ-
   ent biomedical modalities improve task performance, as measured by AUC (area
   under the curve) and accuracy?

2. How effectively do foundation models transfer knowledge across different biomed-
   ical imaging modalities, as evidenced by comparative AUC and accuracy scores in
   cross-modal applications?

Despite the potential benefits, effectively integrating multiple foundation mod-
els presents major challenges. Retraining models on combined data from multiple
modalities demands extensive computational resources and expertise [22]. While
combining pre-trained model embeddings offers an alternative [23,24], the rapidly
growing number of foundation models [25] makes identifying optimal combinations
increasingly complex. Moreover, the lack of standardized frameworks for embedding
fusion hinders our ability to fully exploit these models' collective knowledge.

## 1.2 Contributions

In response to these challenges, we introduce BioFuse, an open-source framework
designed to harness the collective power of diverse biomedical foundation models.
This paper makes the following key contributions:

1. **Enhanced Performance on the MedMNIST+ Benchmark.** Using embeddings generated by BioFuse combined with XGBoost classification, we outperform several existing baselines on the MedMNIST+ benchmark, achieving the highest test AUC in 5 of 12 datasets and best accuracy in 2 datasets. The framework's embeddings maintain near-SOTA performance (within 2% margin) across most remaining datasets, demonstrating the effectiveness of our embedding fusion approach.

2. **Revealing Cross-Modal Capabilities in Biomedical Foundation Models.** Our experiments reveal unexpected cross-modal transfer in both histopathology and radiology models, showcasing the potential for models trained on one imaging modality to generate effective embeddings for others.

3. **An Extensible Framework for Optimised Biomedical Embedding Generation.** BioFuse provides a high-level API that simplifies the generation of task-specific embeddings. It automatically identifies optimal combinations of foundation models and fuses their outputs, addressing the challenges of model integration. The framework's architecture allows for easy incorporation of new models and fusion techniques, ensuring adaptability to emerging developments in biomedical AI.

By providing a standardized approach to embedding fusion and automating model selection, BioFuse enhances the utility of foundation models in biomedicine while opening new avenues for uncovering relationships between diverse biomedical data representations.

## 2. Background and related work

### 2.1 Foundation models in biomedicine

Foundation models, driven by Large Language Models (LLMs), have ushered in a new era in deep learning [26]. Three key factors enabled their development: the Transformer architecture [27] for processing sequential data, advances in GPU capabilities [28], and extensive training datasets [29,30]. These billion-parameter models [12] require substantial computing infrastructure for training [31], but can then solve tasks beyond their original training objectives [32]. While they support zero-shot inference and fine-tuning [26], they are commonly used as feature extractors for classification models [16,33].

Foundation models are typically trained using self-supervised learning (SSL) techniques, including Contrastive Learning [34], Masked Image Modelling (MIM) [35], and self-distillation [36]. These methods create proxy tasks from unlabelled data: contrastive learning differentiates between similar and dissimilar samples, MIM reconstructs masked image regions, and self-distillation leverages the model's own predictions as training targets. By utilizing the inherent structure of data, SSL enables training on much larger datasets without manual annotation. This approach has demonstrated superior performance and transferability compared to traditional supervised learning [34,35,37], making it particularly valuable in medical imaging where labelled data is scarce.

**2.1.1 Unimodal foundation models.** Biomedical imaging foundation models have demonstrated remarkable effectiveness when trained on single modalities (unimodal training). In computational pathology, models like UNI [38], UNI2 [39], Prov-GigaPath [40], and Hibou-B [41] have set new benchmarks through increasingly larger-scale training. UNI outperforms prior models across 34 clinical tasks, while its successor UNI2 expands capabilities to both H&E and Immunohistochemistry (IHC). Prov-GigaPath achieves state-of-the-art performance in cancer subtyping and mutation prediction, and Hibou-B demonstrates high adaptability across pathology applications. In radiology, RAD-DINO [42], pre-trained on a vast corpus of chest X-rays, excels in detecting conditions and capturing detailed features crucial for biomarker discovery and prognosis prediction. Built on architectures like Vision Transformers (ViT) [43] and trained with self-supervised learning, these models demonstrate how large-scale training captures modality-specific nuances and enables superior biomedical imaging performance.

**2.1.2 Vision language models.** Vision-language models (VLMs) integrate visual and textual data to learn joint representations for tasks like image classification, retrieval, and captioning. BioMedCLIP [44] and PubMedCLIP [45] achieve state-of-the-art performance across various biomedical benchmarks, while CheXagent [46] excels in chest X-ray interpretation tasks and CONCH [47] demonstrates superior performance across 13 histopathology benchmarks. These models demonstrate how integrating visual and textual information enhances biomedical imaging analysis.

**2.1.3 Limitations of foundation models.** Despite their capabilities, biomedical foundation models face serious limitations including restricted cross-domain applicability, high computational demands, and challenges in interpretability crucial for clinical decision-making. While using them as feature extractors mitigates the risk of hallucinations [20], effectively leveraging the collective knowledge of multiple specialized models remains an open challenge that could enhance performance and provide more robust biomedical insights.

## 2.2 Related work

Prior work relevant to BioFuse spans three key areas: embedding fusion approaches that combine features from multiple models, cross-modal methods that transfer knowledge between domains, and automated selection techniques that optimize model combinations.

**2.2.1 Embedding fusion.** Recent works have demonstrated the benefits of combining embeddings from multiple models. In histopathology, Neidlinger et al. [24] showed that combining four foundation models improved tumor classification and survival prediction across 13 patient cohorts, but their approach was limited to a single modality and lacked a framework for optimal model selection. Similarly, Zarif et al. [48] and Dong et al. [49] combined two and four pre-trained CNNs respectively for breast and liver cancer classification, but their approaches used bespoke architectures specific to their tasks. In clinical NLP, BioFLAIR [50] combines two embedding models (FLAIR and BioELMo) for biomedical named entity recognition, while the Concatenated BioMed-Transformers [51] fuses three transformer models for medical text classification, but both were limited to single modalities.

**2.2.2 Cross-modal transfer.** Cross-modal approaches have attempted to bridge different modalities in biomedical data. Zipper [52] combines two pre-trained unimodal models for speech recognition and text-to-speech tasks using multi-tower decoders with cross-attention, but its computational complexity limits practical application. BioBridge [53] uses knowledge graphs to connect two unimodal foundation models for protein sequence-text retrieval and drug design tasks, yet fails to fully exploit the embedded knowledge across different biomedical domains.

**2.2.3 Automated model selection.** In automated model selection, ACE [54] uses reinforcement learning to select optimal combinations of word embedding models for NLP tasks like named entity recognition and part-of-speech tagging. However, its focus on text-only tasks fails to address the multimodal nature of biomedical data, and its computational demands make it impractical for high-dimensional data where rapid model selection is often needed.

**2.2.4 Limitations of related work.** While these approaches have made impressive strides, they often focus on specific domains or tasks, lacking the versatility to leverage image extraction capabilities across models trained on different modalities. Moreover, the untapped potential in combining foundation models to enhance performance and generalisability across diverse modalities remains a non-trivial challenge. BioFuse aims to address these limitations by providing a flexible framework that can integrate multiple foundation models across various biomedical domains, facilitating both embedding fusion and cross-modal transfer while automating the selection of optimal model combinations.

## 3. Methods

### 3.1 Ethics statement

This research was conducted under ethical approval from the institutional review board (approval code CS17485).

## 3.2 BioFuse framework

**3.2.1 Overview.** BioFuse is an open-source framework that automates the selection, extraction, and fusion of embeddings from multiple pre-trained biomedical foundation models across diverse modalities such as radiographs, histopathology slides, and clinical text. Its modular design supports seamless integration of new models and fusion methods, keeping pace with rapid advancements.

BioFuse's primary goal is to generate optimal embeddings by leveraging multiple models via vector concatenation. These fused embeddings encapsulate multimodal information in a unified format. To assess their quality, BioFuse employs an approach similar to linear probing [55], but with a more sophisticated classifier. Specifically, we use XGBoost [56], a powerful tree-based machine learning algorithm, to train on the frozen, fused embeddings for various downstream tasks. This evaluation method provides a reliable measure of embedding quality without requiring computationally expensive fine-tuning of the foundation models themselves.

Users can utilise these embeddings as input features for custom models tailored to specific research questions or clinical applications. By offering optimised embeddings and demonstrating their effectiveness through advanced probing, BioFuse serves as a versatile feature extraction framework for biomedical imaging tasks.

**3.2.2 System architecture.** BioFuse's architecture comprises three components: pre-trained foundation models, the BioFuseModel, and the search module (see Fig 1):

1. **Pre-trained Foundation Models.** Existing, pre-trained models from various biomedical domains are loaded without fine-tuning to preserve their original capabilities.

2. **BioFuseModel.** The core component that processes and integrates embeddings from multiple pre-trained models in a single forward pass per input. It preprocesses each model's input, extracts embeddings, and fuses them via vector concatenation, preserving the strengths of each model while ensuring efficiency.

3. **Search Module.** Evaluates embedding performance from different model combinations (each represented by a BioFuseModel) by computing validation accuracy using XGBoost as a lightweight downstream classifier. This step automates identifying optimal configurations.

**3.2.3 Foundation models in BioFuse.** Foundation models in BioFuse were selected based on architectural diversity, performance, efficiency, and accessibility. Both unimodal and vision-language models (VLMs) are included to capture diverse biomedical information, with VLMs chosen for their pre-training on paired text-image data. Models trained on high-quality biomedical datasets in various imaging modalities, from macroscopic radiological images to microscopic histological data, were preferred for a comprehensive representation of knowledge.

The selection process prioritized models documented in peer-reviewed literature with demonstrated excellence across multiple biomedical applications. We carefully balanced computational efficiency against performance to ensure BioFuse remains practical for real-world implementation. Additionally, we favored models with standardized implementations on platforms such as Hugging Face [57] to enhance accessibility and community support. Through this thoughtful curation, BioFuse incorporates a diverse array of foundation models that collectively provide robust and adaptable capabilities across the biomedical domain.

Table 1 summarizes the foundation models supported by BioFuse, including dataset size, pre-training methods, and encoder types.

**3.2.4 Embedding extraction.** BioFuse streamlines the extraction of embeddings from diverse pre-trained foundation models across multiple biomedical imaging modalities. The framework leverages established libraries including Hugging Face Transformers [59], OpenCLIP [60], and PyTorch Image Models [61] to facilitate this process. The system automatically handles model and preprocessor loading while applying appropriate model-specific preprocessing—such as image resizing and normalization—before generating embeddings.

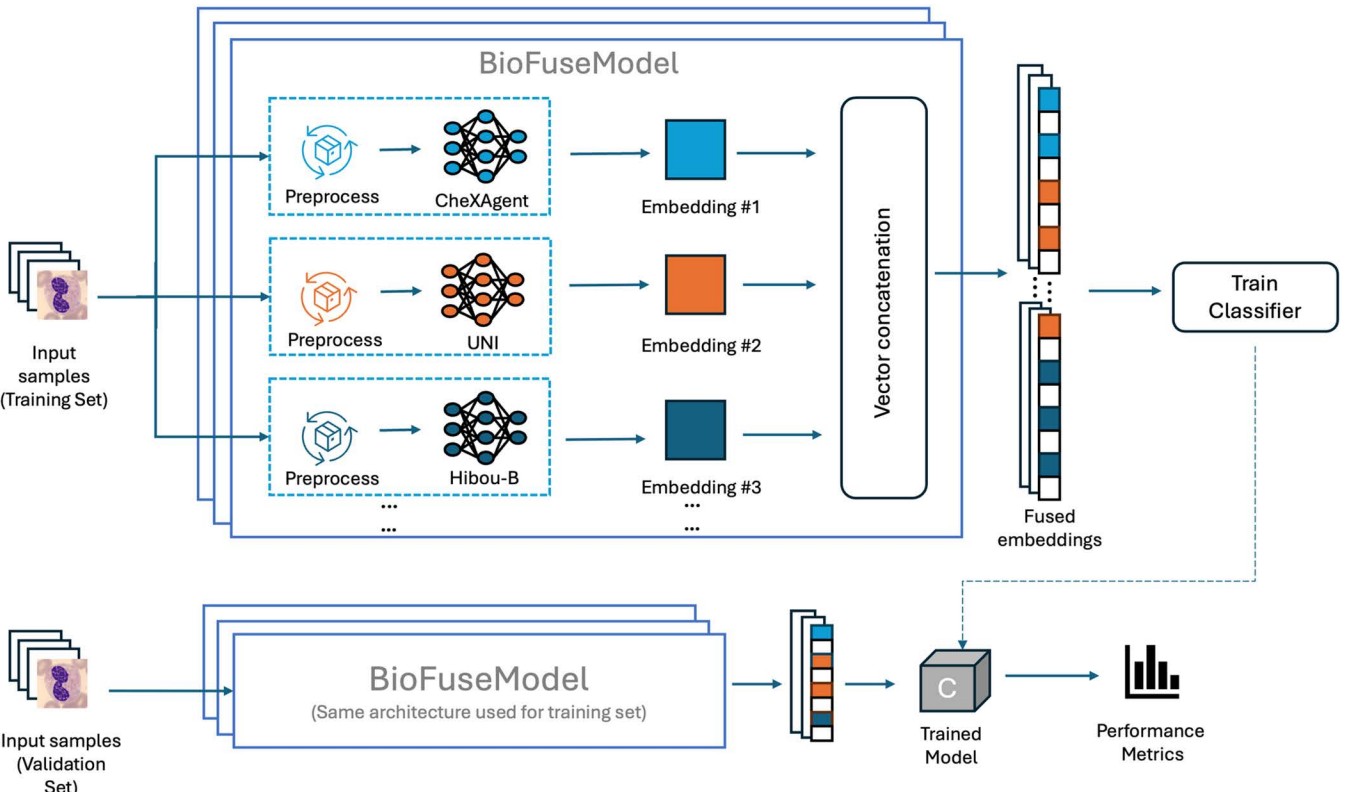

**Fig 1. BioFuse architecture and workflow.** The upper section illustrates the training process: input samples are preprocessed and fed into multiple foundation models to generate embeddings, which are then concatenated and used to train a classifier. The lower section shows the evaluation process using the same BioFuseModel architecture on a validation set, followed by performance assessment of the trained model.

**Table 1. Summary of foundation models supported in BioFuse.**

| Model | Type | Dataset | Dataset size | Pre-training method | Text Encoder | Vision Encoder |
|---|---|---|---|---|---|---|
| BioMedCLIP [44] | VLM | Figure-caption pairs from PubMed Central | 15M | Contrastive Learning | PubMedBERT | ViT |
| PubMedCLIP [45] | VLM | Radiology Objects in Context (ROCO) | 81K | Contrastive Learning | Transformer | ViT |
| CheXagent [46] | VLM | CheXinstruct (28 CXR datasets) | 6.1M I-Q-A, 1.1M CXRs | Instruction tuning [58], Vision-language pre-training | Custom clinical LLM | ViT |
| CONCH [47] | VLM | Histology slides | 1.17M image-caption pairs | Vision-Language pre-training [32] | CLIP-based | ViT-B/12 |
| rad-dino [42] | Unimodal | MIMIC-CXR, CheXpert, NIH-CHR, PadChest, BRAX | 838K | DINOv2 | – | ViT |
| UNI [38] | Unimodal | Mass-100K | 100M tiles, 100K WSIs | DINOv2 | – | ViT-L/16 |
| UNI2 [39] | Unimodal | 200M H&E and IHC images | 200M tiles, 350K WSIs | DINOv2 | – | ViT-h/14-reg8 |
| Hibou-B [41] | Unimodal | Histology slides | 512M tiles, 1M WSIs | DINOv2 | – | ViT-B/14 |
| Prov-GigaPath [40] | Unimodal | Histology slides | 1.3B tiles, 171K WSIs | DINOv2 | – | ViT |

VLM: Vision-Language Model, ViT: Vision Transformer, I-Q-A: Instruction-Question-Answer.

The framework employs GPU acceleration to enhance inference speed when processing large-scale datasets. Memory management is optimised through immediate release of intermediate outputs after use, enabling the system to scale effectively with increasingly complex inputs. These technical enhancements ensure BioFuse can efficiently handle diverse biomedical imaging modalities while maintaining computational efficiency in resource-constrained environments.

**3.2.5 Fusion methodology.** BioFuse fuses embeddings from multiple pre-trained models using **vector concatenation**, combining embeddings into a single representation.

Let $e_i \in \mathbb{R}^{d_i}$ be the embedding from the $i$-th model, where $d_i$ is its dimensionality. Given $n$ models, the fused embedding is:

$$e_{concat} = [e_1; e_2; \ldots; e_n] \tag{1}$$

where $[;]$ denotes concatenation along the feature dimension, yielding a total size:

$$d_{concat} = \sum_{i=1}^{n} d_i \tag{2}$$

For instance, if $e_1 \in \mathbb{R}^{512}$ and $e_2 \in \mathbb{R}^{768}$, then $e_{concat} \in \mathbb{R}^{1280}$.

**Advantages of vector concatenation:**

- Efficient & Simple: No additional parameters or projection layers, reducing computational complexity.

- Feature Preservation: Retains all modality-specific features without loss.

- Scalable: New models can be seamlessly integrated without retraining fusion components.

- Interpretable: Maintains per-model feature separation, aiding downstream classifiers.

**Feature-scale alignment.** Although the nine foundation models differ in objective, domain, and ViT scale, the length of their flattened embeddings is still within a single order of magnitude (512–1536 dimensions). Any scale mismatch across those vectors could, in principle, bias the downstream learner. Two properties mitigate this risk: (i) concatenation keeps each model's features in isolated sub-blocks, so no cross-model normalisation is required; (ii) the downstream classifier is XGBoost, a tree-based ensemble whose splitting decisions depend primarily on the relative ordering of feature values rather than their absolute magnitudes [56,62]. While feature importance scores may retain some scale sensitivity, empirical studies on gradient-boosted trees demonstrate that standard scaling transformations (z-score, min–max) typically change test accuracy by less than 0.5 percentage points [63], indicating that scale heterogeneity is not a limiting factor for BioFuse's performance.

**3.2.6 Automated selection of models.** BioFuse automates model selection, identifying the optimal set for each task without manual intervention. It evaluates all $2^n - 1$ model combinations, leveraging GPU-accelerated XGBoost to efficiently search this exponential space. Embeddings are fused and rapidly assessed using gradient boosting. GPU acceleration mitigates computational overhead by parallelizing tree construction and handling large feature matrices efficiently. This enables BioFuse to evaluate model combinations in seconds and process larger datasets within minutes, ensuring thorough exploration of the combinatorial space.

An on-disk cache stores previously generated embeddings to avoid redundant computations. BioFuse selects the combination with the highest validation accuracy, ensuring embeddings are tailored to the specific task while maintaining generalisation within the dataset.

## 3.3 Dataset

As Illustrated in Fig 2, our study uses 12 2D datasets from the MedMNIST+ benchmark [64], a large-scale, MNIST-like collection of standardized biomedical images designed for various machine learning tasks in the medical domain. The

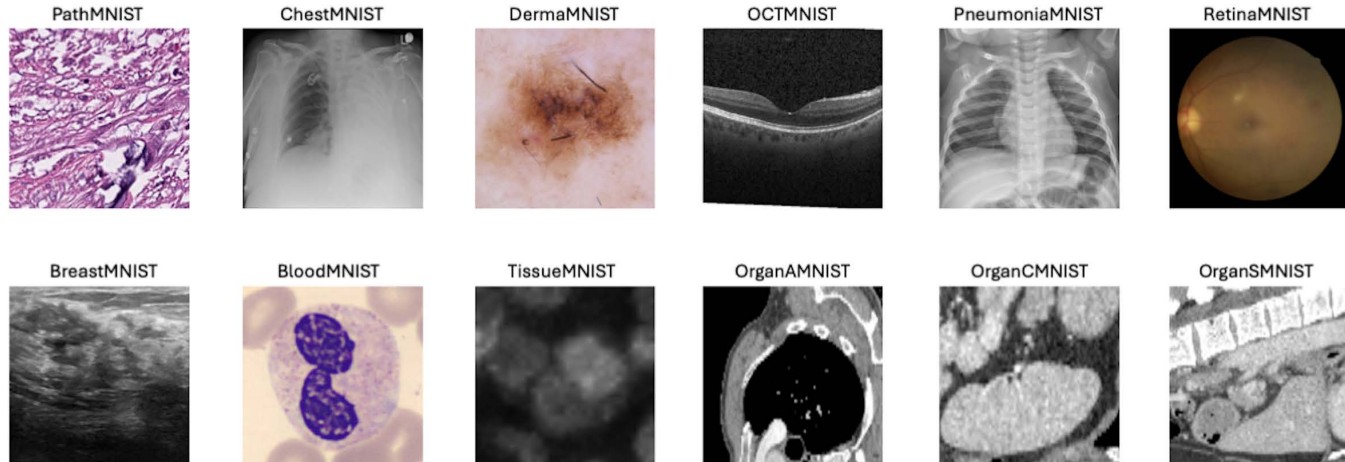

**Fig 2. Visual overview of the MedMNIST+2D datasets.** Sample images at 224x224 resolution showcase the diverse collection spanning multiple medical imaging modalities, illustrating the wide range of diagnostic tasks represented in the benchmark.

dataset was accessed periodically between June 1, 2024, and February 28, 2025, for research purposes. Key features of the dataset include:

1. Multi-modal: MedMNIST+ covers a wide range of biomedical imaging modalities, including X-Ray, OCT, Ultrasound, CT, Histopathology and Electron Microscopy images.

2. Standardized: All images are pre-processed into a uniform $224 \times 224$ resolution for 2D datasets, with corresponding classification labels. The datasets also come with consistent train-validation-test splits.

3. Multi-task: The dataset supports various machine learning tasks, including binary and multi-class classification, ordinal regression, and multi-label classification.

4. Multi-scale: MedMNIST includes approximately 708K 2D images, with dataset sizes ranging from 780 to 236,386 samples.

The decision to focus on 2D datasets was driven by the use of a simple classifier which is well-suited for 2D image analysis tasks. Table 2 provides an overview of the MedMNIST2D datasets used in this study.

MedMNIST+ enables a comprehensive evaluation of BioFuse across multi-class, binary, and multi-label classification, spanning diverse dataset sizes and class distributions. For consistency, we treat the ordinal regression task (RetinaM-NIST) as a multi-class classification problem, a standard approach for handling ordinal data in machine learning [74].

### 3.4 Experiments

**3.4.1 Objectives.** Our experiments assess:

1. Performance improvement: Evaluate whether fusing embeddings via vector concatenation outperforms individual models on MedMNIST+ using AUC and Accuracy.

2. Cross-modal transfer: Test if features from one modality (e.g., histopathology) enhance performance when combined with another (e.g., radiographs), measured by AUC and Accuracy.

**Table 2. Overview of MedMNIST+2D datasets.**

| Dataset | Data Modality | Tasks(# Classes) | # Samples | # Train/Valid/Test |
|---------|---------------|------------------|-----------|--------------------|
| PathMNIST [65] | Colon Pathology | Multi-Class (9) | 107,180 | 89,996/10,004/7,180 |
| ChestMNIST [66] | Chest X-Ray | Multi-Label (14) Binary-Class (2) | 112,120 | 78,468/11,219/22,433 |
| DermaMNIST [67] | Dermatoscope | Multi-Class (7) | 10,015 | 7,007/1,003/2,005 |
| OCTMNIST [68] | Retinal OCT | Multi-Class (4) | 109,309 | 97,477/10,832/1,000 |
| PneumoniaMNIST [68] | Chest X-Ray | Binary-Class (2) | 5,856 | 4,708/524/624 |
| RetinaMNIST [69] | Fundus Camera | Ordinal Regression (5) | 1,600 | 1,080/120/400 |
| BreastMNIST [70] | Breast Ultrasound | Binary-Class (2) | 780 | 546/78/156 |
| BloodMNIST [71] | Blood Cell Microscope | Multi-Class (8) | 17,092 | 11,959/1,712/3,421 |
| TissueMNIST [72] | Kidney Cortex Microscope | Multi-Class (8) | 236,386 | 165,466/23,640/47,280 |
| OrganAMNIST [73] | Abdominal CT | Multi-Class (11) | 58,830 | 34,561/6,491/17,778 |
| OrganCMNIST [73] | Abdominal CT | Multi-Class (11) | 23,583 | 12,975/2,392/8,216 |
| OrganSMNIST [73] | Abdominal CT | Multi-Class (11) | 25,211 | 13,932/2,452/8,827 |

Data sourced from [64].

3. Generalisation: Assess BioFuse's ability to maintain strong performance across diverse biomedical domains using nine foundation models.

4. Fusion method validation: Compare vector concatenation against alternative fusion strategies (self-attention) and single-model baselines to justify the chosen approach.

5. Robustness evaluation: Assess BioFuse's performance under realistic image corruptions using MedMNIST-C.

These objectives establish BioFuse as a versatile framework for multimodal biomedical imaging, leveraging complementary information to improve classification performance across datasets.

**3.4.2 Hardware configuration.** The experiments were conducted on a server with an NVIDIA A100 GPU (80GB VRAM), an AMD EPYC 7713 CPU (64-core, 128-thread CPU), and 1 TB RAM. While the CPU and memory were shared, the GPU was dedicated to BioFuse, ensuring uninterrupted computation for embedding extraction, model fusion, and classification.

**3.4.3 Model selection classifier.** For internal evaluation (see Fig 1), we used XGBoost [56], a scalable gradient-boosted decision tree algorithm known for its efficiency in binary and multi-class classification. The model was trained on fused embeddings from multiple foundation models.

Although embedding order can influence performance, we found this effect negligible. Optimizing order introduces substantial complexity—$2^n - 1$ model combinations and $n!$ possible orders per combination—making exhaustive search impractical. To maintain computational tractability, we fixed the concatenation order.

Hyperparameters were manually selected following [75], balancing efficiency and accuracy. We set `n_estimators=250`, `learning_rate=0.1`, and `max_depth=6` to prevent overfitting while capturing biomedical data patterns. The objective function was `multi:softprob` for multi-class, `binary:logistic` for binary classification, and a one-versus-rest strategy for multi-label tasks. Training used XGBoost's GPU acceleration to reduce computation time.

**3.4.4 Training and evaluation workflow.** Our evaluation of BioFuse follows a structured workflow, as illustrated in Fig 3:

1. **Dataset Preparation:** BioFuse is provided with training and validation sets for each dataset in MedMNIST+2D.

2. **Model Selection:** It identifies the optimal combination of foundation models for each dataset (see Fig 1).

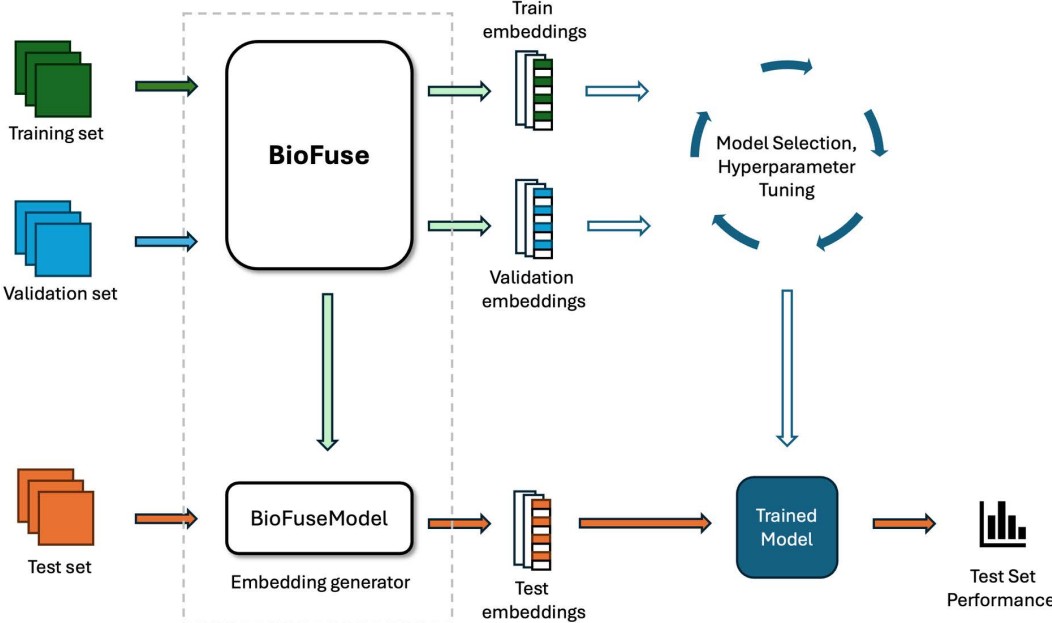

**Fig 3. High-level workflow of BioFuse within a typical machine learning pipeline.** BioFuse serves as a sophisticated embedding generator, accepting training and validation sets as inputs. As outputs, BioFuse provides embeddings for both the training and validation sets, along with a configured BioFuseModel. This BioFuseModel acts as an embedding generator for subsequent use on the test set.

3. **Embedding Generation:** Using the selected model combination, BioFuse generates training and validation embeddings and returns a configured BioFuseModel (embedding generator).

4. **Classifier Selection and Tuning:** For the final evaluation, we use XGBoost, tuning its hyperparameters on training embeddings and selecting the configuration that achieves the highest validation accuracy.

5. **Test Embedding Generation:** The trained BioFuseModel generates embeddings for the test set, ensuring consistency with the training and validation process.

6. **Final Evaluation:** The tuned XGBoost model (from Step 4) is evaluated on the test set embeddings to assess overall performance.

To ensure a comprehensive final evaluation on the test sets, we conducted independent hyperparameter tuning for each dataset in the MedMNIST+ benchmark (as described in Step 4). The XGBoost hyperparameter search range is detailed in S2 Table.

Performance was measured using test AUC and accuracy, with results averaged over three independent runs to enhance robustness.

**3.4.5 Evaluation metrics.** We employ two complementary metrics in our evaluation framework:

**Accuracy** is used for internal model selection during BioFuse's automated search process:

$$\text{Accuracy} = \frac{\text{Number of Correct Predictions}}{\text{Total Number of Predictions}}$$

**Area Under the ROC Curve (AUC-ROC)** serves as our primary reporting metric for final test set performance. This choice aligns with MedMNIST+ benchmarks and provides a threshold-independent assessment of model quality.

AUC-ROC represents the probability that a randomly chosen positive example ranks higher than a negative one [76], with values ranging from 0.5 (random performance) to 1.0 (perfect classification).

For comprehensive comparison, we report both AUC-ROC and accuracy in our final results tables.

## 4. Results

### 4.1 Overview

On the MedMNIST+ benchmark, BioFuse outperforms several existing methods, achieving the highest test AUC on 5 of 12 datasets and near-SOTA performance (within 2% margin) on the remaining datasets. It also achieves the best test accuracy on 2 of 12 datasets and near-SOTA accuracy on another 4 (see Table 3). These results highlight the effectiveness of embedding fusion from diverse foundation models in improving performance across a broad range of biomedical image classification tasks. The exact XGBoost parameters are listed in S3 Table.

A particularly interesting finding is the versatility of histopathology and radiology foundation models. Histopathology models UNI, Hibou-B, CONCH and Prov-GigaPath and radiology models CheXagent and rad-dino consistently appear in the top-performing model combinations, even when applied to modalities different from their pre-training data. This cross-modality effectiveness suggests these models learn generalisable features useful across various biomedical imaging tasks.

### 4.2 Cross-Modal performance

**4.2.1 Transfer to other medical imaging modalities.** Figure 4 (AUC-ROC) and Fig 5 (accuracy) list the top-three single-model scores per dataset. Hyperparameters for XGBoost were frozen (see 3.4 Experiments). CLIP [32] remains our non-medical baseline.
**Fixed hyperparameters in single-model heatmaps.** For the single-model heatmaps, we use a fixed XGBoost configuration across all experiments. Running a full Bayesian search for every (dataset, backbone) pair would require

**Table 3. Performance comparison between BioFuse and best existing methods on MedMNIST+; bold values indicate the best performance for each dataset.**

| Dataset | BioFuse | | Best Other Method | | Improvement | | BioFuse Model Combination |
|---|---|---|---|---|---|---|---|
| | AUC | ACC | AUC | ACC | AUC | ACC | |
| BloodMNIST | **0.999** | 0.988 | 0.998[b] | **0.990**[a] | +0.001 | −0.002 | UN,PG,HB,CA |
| BreastMNIST | **0.944** | 0.878 | 0.939[b] | **0.906**[c] | +0.005 | −0.028 | CO,CA |
| DermaMNIST | **0.970** | **0.855** | 0.968[d] | 0.847[a] | +0.002 | +0.008 | BC,UN,HB,CA |
| OCTMNIST | **0.992** | 0.798 | **0.992**[a] | **0.903**[d] | +0.000 | −0.105 | BC,PC,RD,PG,HB,CA,UN2 |
| OrganAMNIST | 0.996 | 0.920 | **0.999**[d] | **0.969**[d] | −0.003 | −0.049 | BC,PC,CO,PG,HB,UN2 |
| OrganCMNIST | 0.993 | 0.883 | **0.997**[d] | **0.992**[e] | −0.004 | −0.109 | BC,PC,CO,UN,PG,HB,CA,UN2 |
| PathMNIST | 0.994 | **0.972** | **0.997**[a] | 0.961[d] | −0.003 | +0.011 | BC,UN,HB,CA,UN2 |
| PneumoniaMNIST | 0.994 | 0.930 | 0.995[g] | **0.961**[g] | −0.001 | −0.031 | BC,UN,PG,CA |
| RetinaMNIST | 0.860 | 0.638 | **0.878**[e] | **0.641**[e] | −0.018 | −0.003 | PC,CO,RD,UN,UN2 |
| TissueMNIST | 0.933 | 0.693 | **0.953**[c] | **0.742**[c] | −0.020 | −0.049 | CO,RD,UN,PG,HB,CA,UN2 |
| OrganSMNIST | 0.979 | 0.789 | **0.985**[d] | **0.834**[a] | −0.006 | −0.045 | BC,PC,RD,UN,PG,HB |
| ChestMNIST | **0.835** | 0.949 | 0.822[a] | **0.959**[h] | +0.013 | −0.010 | BC,PC,RD,UN,CA |

**Model Legend:** BC: BioMedCLIP, PC: PubMedCLIP, CO: CONCH, RD: rad-dino, UN: UNI, PG: Prov-GigaPath, HB: Hibou-B, CA: CheXagent, UN2: UNI2.

**Reference Models:** [a]DenseNet-121 [77], [b]ResNet-18 (224) [64], [c]AlexNet [77], [d]DINO ViT-B/16 [77], [e]VGG16 [77], [f]MEDVIT-S [78], [g]MEDVIT-L [78].

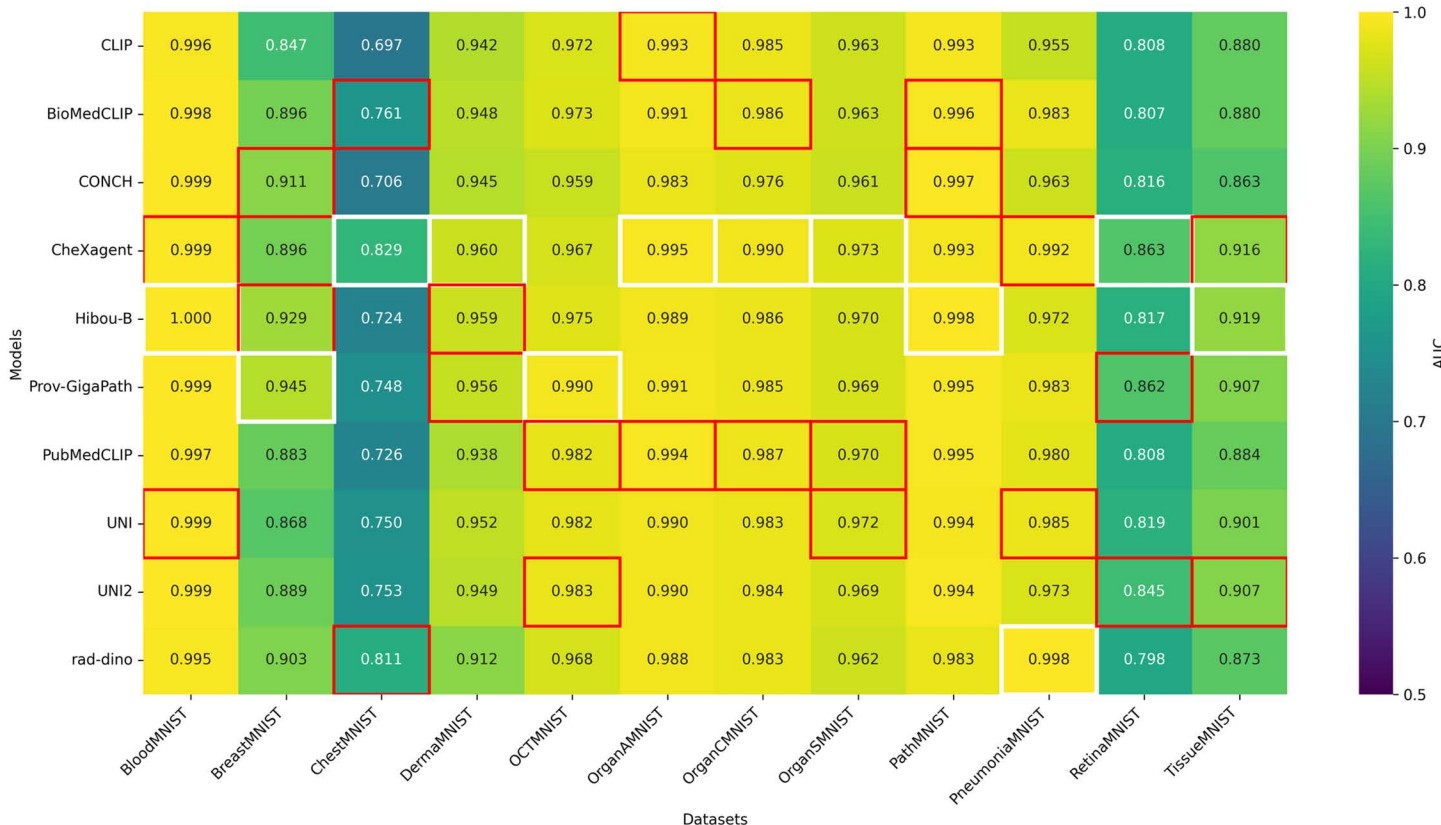

**Fig 4. Heatmap of test AUC, showing single model performance across MedMNIST+ datasets.** CLIP is included as a baseline general-purpose foundation model for comparison. White box indicates the top-performer while red boxes highlight the 2nd and 3rd top performers for each dataset; More than three models may be highlighted if they share identical values.

$9 \times 12 \times 100$ trials $\approx 10,800$ additional fits. Extending this to all backbone combinations would increase the count by orders of magnitude. Since the heatmaps aim to rank the relative quality of individual backbones, a shared configuration suffices and isolates performance differences to the backbone itself. For fusion experiments (concatenation, self-attention, oracle-single), we perform full Bayesian search as only one backbone set per dataset requires tuning, keeping the computational budget manageable.

**Radiology → Histopathology.** Models whose pre-training focus is radiology demonstrate strong transfer to histopathology tasks:

- CheXagent (*CA*)—originally trained on chest X-rays—achieves first place in both metrics on *DermaMNIST* and places second on *TissueMNIST* for both accuracy and AUC.

**Radiology → Microscopy/Ophthalmology.** Radiology models also transfer effectively to other imaging modalities:

- CheXagent (*CA*) achieves first place on *RetinaMNIST* (first in both metrics) and reaches third place on *BloodMNIST* for accuracy (0.974) and tied second place for AUC (0.999).

**Histopathology → Radiology.** Conversely, histopathology models also transfer effectively to radiology benchmarks:

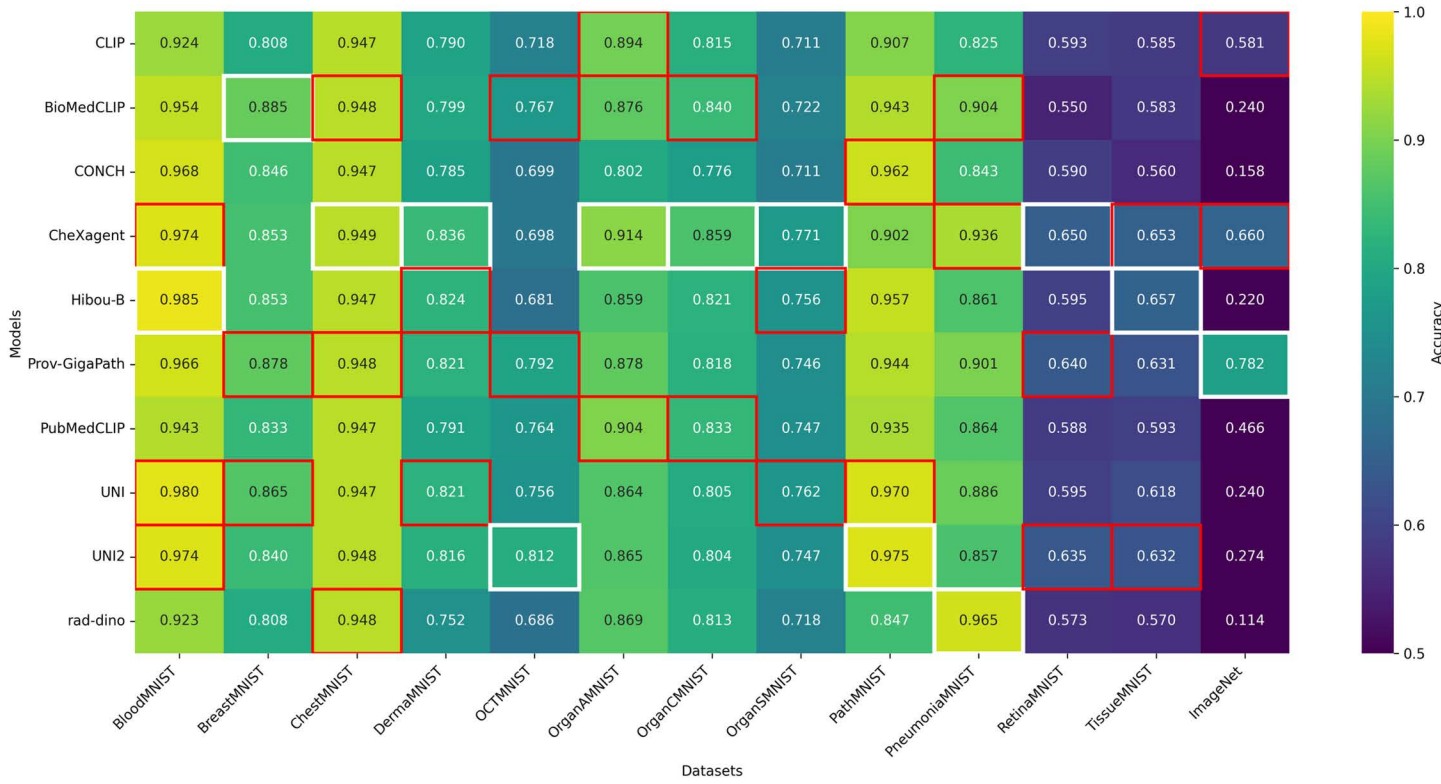

**Fig 5. Heatmap of test accuracy, showing single model performance across MedMNIST+ datasets and ImageNet-1K (top-1).** CLIP is included as a baseline general-purpose foundation model for comparison. White boxes indicate top performers, while red boxes highlight second and third best performers for each dataset. When multiple models achieve identical performance, more than three boxes may be highlighted.

- Prov-GigaPath (*PG*) achieves first in AUC (0.945) and second in accuracy (0.878) on *BreastMNIST*, while placing tied second on *ChestMNIST* accuracy.

- Hibou-B (*HB*) ranks second on *BreastMNIST* AUC (0.929) and third on *OrganSMNIST* accuracy (0.756).

- UNI (*UN*) enters the top-three on *BreastMNIST* (third accuracy), *OrganSMNIST* (second in both metrics), and *PneumoniaMNIST* (third AUC).

- CONCH (*CO*) secures third place on *BreastMNIST* AUC (0.911).

**Histopathology → Microscopy/Ophthalmology.** Histopathology models also excel on microscopy and ophthalmology tasks:

- Hibou-B (*HB*) achieves first place on *BloodMNIST* with perfect AUC (1.000) and top accuracy (0.985).

- UNI and UNI2 place second and third, respectively, on *BloodMNIST* accuracy, with UNI also second in AUC. UNI2 reaches third place on *RetinaMNIST* for both metrics.

- Prov-GigaPath (*PG*) ranks second on *RetinaMNIST* for both accuracy (0.640) and AUC (0.862).

**Baseline.** CLIP lags behind specialist medical models but still reaches the top-three on *OrganAMNIST* for both accuracy (0.894) and AUC (0.993).

Taken together, these heatmaps demonstrate genuine multidirectional transfer: radiology-centric models can excel on histopathology and ophthalmology tasks, while histopathology models perform competitively on radiology and microscopy benchmarks. No single model dominates every modality—hence the motivation for the fusion strategy deployed in BioFuse.

### 4.3 Transfer to natural images

To comprehensively evaluate the generalisation capabilities of medical foundation models, we assess their performance on the ImageNet-1K [79] test set, a benchmark dataset widely used in computer vision. While we don't expect medical models to match the performance of models trained specifically on natural images, their performance provides valuable insights into their general visual understanding capabilities. Surprisingly, some medical models demonstrate remarkable transfer ability, with Prov-GigaPath and CheXagent achieving 78.2% and 66.0% top-1 accuracy respectively, outperforming CLIP (58.1%) despite CLIP being trained on 400M natural image-text pairs. Table 4 presents the complete ImageNet-1K test set performance results.

### 4.4 Ablation studies

**4.4.1 Ablation 1: Self-attention fusion.** To test whether a learnable fusion layer can outperform the simple concatenation used in BioFuse, we replaced concatenation with a *multi-head self-attention* (MHSA) block applied across the per-model embeddings (attention formulation as in [27]). Following common practice ($d_{head}$ = 64[43]), we swept over 4, 8, and 12 heads. The projected dimension is therefore *proj_dim* = $d_{head} \times n_{heads} \in \{256, 512, 768\}$. For each dataset we selected the best-performing projection size (column *proj_dim* in Table 5) and fed those embeddings into the same XGBoost hyper-parameter search that was used for the concatenation baseline.

**Discussion.** Self-attention (SA) fusion helps only two sets—*OCTMNIST* (+1.7 pp ACC) and *PathMNIST* (+0.4 pp AUC)—and trails concatenation elsewhere. The largest deficits occur on *OrganCMNIST* (–26 pp AUC, –52 pp ACC) and *OrganSMNIST* (–1 pp AUC, –3.8 pp ACC). We identify three interacting factors:

1. Extra parameters vs. class-wise sample size. SA introduces a substantial number of new weights, which are difficult to learn reliably when each class provides only a few hundred training images (as in OrganC/S). Parameter-free concatenation avoids this over-fitting risk.

2. Radiology–histology model mix. The BioFuse pool combines radiology backbones (RD, CA) with histology-centric ones (HB, UN, UN2, PG, CO). When the target domain is a radiology CT variant such as OrganC/S, features from the

**Table 4. Performance comparison of biomedical foundation models on ImageNet-1K. Bold values indicate best performance.**

| Model | Top-1 ACC | Top-5 ACC | Pre-training Dataset Size |
|---|---|---|---|
| CoCa | **0.910** | — | 4.8B (3B labeled images + 1.8B image-text pairs) |
| Prov-GigaPath | 0.782 | 0.944 | 1.3B tiles (171K WSIs) |
| CheXAgent | 0.660 | 0.908 | 6.1M I-Q-A triplets (1.1M CXRs) |
| CLIP | 0.581 | 0.870 | 400M image-text pairs |
| PubMedCLIP | 0.466 | 0.772 | 81K image-text pairs |
| UNI2 | 0.274 | 0.517 | 100M tiles (100K WSIs) |
| BioMedCLIP | 0.240 | 0.502 | 15M image-text pairs |
| UNI | 0.240 | 0.479 | 100M tiles (100K WSIs) |
| Hibou-B | 0.220 | 0.444 | 512M tiles (1M WSIs) |
| CONCH | 0.158 | 0.347 | 1.17M image-caption pairs |
| rad-dino | 0.114 | 0.268 | 838K images |

**Table 5. Ablation 1 – Self-attention fusion vs. concatenation.** The "Concat" columns reproduce the BioFuse baseline; the *Model combination* column lists the backbones selected for the *self-attention* run only—the corresponding concat combinations are given in Table 3. Δ = (Self-attn – Concat). Backbones printed in bold overlap with the concat baseline.

| Dataset | Concat | | Self-attn | | Δ | | Proj | Model combination |
|---|---|---|---|---|---|---|---|---|
| | AUC | ACC | AUC | ACC | AUC | ACC | dim | |
| BloodMNIST | 0.999 | 0.988 | 0.999 | 0.982 | +0.000 | −0.006 | 768 | CO, UN, **HB, CA** |
| BreastMNIST | 0.944 | 0.878 | 0.916 | 0.872 | −0.028 | −0.006 | 512 | BC, PC, RD, PG, HB, **CA** |
| DermaMNIST | 0.970 | 0.855 | 0.963 | 0.840 | −0.007 | −0.015 | 768 | **CA** |
| OCTMNIST | 0.992 | 0.798 | 0.990 | 0.815 | −0.002 | **+0.017** | 768 | **PC, RD, PG, CA, UN2** |
| OrganAMNIST | 0.996 | 0.920 | 0.990 | 0.873 | −0.006 | −0.047 | 768 | **BC, PC,** RD, UN, **PG,** CA, **UN2** |
| OrganCMNIST | 0.993 | 0.883 | 0.730 | 0.366 | −0.263 | −0.517 | 768 | RD, **CA, UN2** |
| PathMNIST | 0.994 | 0.972 | 0.998 | 0.955 | **+0.004** | −0.017 | 768 | RD, **HB, CA, UN2** |
| PneumoniaMNIST | 0.994 | 0.930 | 0.994 | 0.923 | +0.000 | −0.007 | 512 | RD, HB, UN2 |
| RetinaMNIST | 0.860 | 0.638 | 0.812 | 0.595 | −0.048 | −0.043 | 768 | **RD, UN** |
| TissueMNIST | 0.933 | 0.693 | 0.926 | 0.678 | −0.007 | −0.015 | 768 | **HB, CA** |
| OrganSMNIST | 0.979 | 0.789 | 0.968 | 0.751 | −0.011 | −0.038 | 768 | **BC,** CO, **UN, PG, HB** |
| ChestMNIST | 0.835 | 0.949 | 0.831 | 0.949 | −0.004 | 0.000 | 768 | **PC, RD, CA** |

histology models are often weak or misaligned. Concatenation leaves these channels untouched; SA, however, learns cross-model attention weights that can *amplify* conflicting cues, sharply reducing accuracy.

3. Feature coherence across models. The two datasets that improve (*OCTMNIST*, *PathMNIST*) have strong texture cues extracted consistently by all backbones, allowing SA to reinforce aligned signals and yield modest gains.

Given this trade-off and the added tuning cost, plain concatenation remains the recommended fusion strategy for BioFuse under the current settings. The exact XGBoost parameters for SA are listed in S4 Table. Future work will explore sparsity-constrained or modality-aware attention to curb overfitting when radiology and histology embeddings diverge.

**4.4.2 Ablation 2: Oracle-single.** In normal BioFuse training we brute-force all backbone combinations, pick the set that gives the highest validation accuracy, and then tune XGBoost on that fused feature space. For the "oracle-single" we restrict the search to individual backbones only. The single model that tops the validation leaderboard for each dataset is treated as the oracle, tuned with the same XGBoost procedure, and evaluated on the test set.

S5 Table lists the tuned hyper-parameters; Table 6 compares oracle-single performance with the concatenation baseline.

**Discussion.** Using only the single best backbone generally underperforms concatenation (both metrics decrease on eight of twelve datasets). A modest AUC gain is observed on *PathMNIST*, but large drops occur on *OrganCMNIST* and *OrganSMNIST*, confirming that fusion captures complementary information not present in any single model. Consequently, BioFuse's concatenation scheme remains the strongest overall configuration.

### 4.5 Robustness to realistic image corruptions

**Benchmark.** We assess robustness with **MedMNIST-C** [80], which adds eleven medically motivated corruptions (five severity levels each) to every MedMNIST+ dataset, mirroring ImageNet-C [81]. Unless stated otherwise, we evaluate the BioFuse *concatenation* baseline—namely the best backbone combination for each dataset listed in Table 3—trained only on the clean MedMNIST+ data; corrupted images are simply resized to $224 \times 224$, and no further fine-tuning is performed. All metrics are computed with the official medmnistc-api toolkit [82].

**Metrics.** MedMNIST-C reports three error-based quantities (lower is better) (Table 7):

**Table 6. Ablation 2 – Oracle-single vs. concatenation.** Concat figures are the main BioFuse results; Oracle-single uses only the best individual model for each dataset. Δ = Oracle − Concat.

| Dataset | Concat | | Oracle-single | | Δ | | Best model |
|---|---|---|---|---|---|---|---|
| | AUC | ACC | AUC | ACC | AUC | ACC | |
| BloodMNIST | 0.999 | 0.988 | 0.999 | 0.985 | +0.000 | −0.003 | HB |
| BreastMNIST | 0.944 | 0.878 | 0.913 | 0.853 | −0.031 | −0.025 | CO |
| DermaMNIST | 0.970 | 0.855 | 0.965 | 0.849 | −0.005 | −0.006 | CA |
| OCTMNIST | 0.992 | 0.798 | 0.974 | 0.702 | −0.018 | −0.096 | CA |
| OrganAMNIST | 0.996 | 0.920 | 0.995 | 0.912 | −0.001 | −0.008 | PC |
| OrganCMNIST | 0.993 | 0.883 | 0.755 | 0.385 | −0.238 | −0.498 | CA |
| PathMNIST | 0.994 | 0.972 | 0.998 | 0.957 | **+0.004** | −0.015 | HB |
| PneumoniaMNIST | 0.994 | 0.930 | 0.992 | 0.936 | −0.002 | +0.006 | CA |
| RetinaMNIST | 0.860 | 0.638 | 0.816 | 0.590 | −0.044 | −0.048 | HB |
| TissueMNIST | 0.933 | 0.693 | 0.924 | 0.674 | −0.009 | −0.019 | HB |
| OrganSMNIST | 0.979 | 0.789 | 0.671 | 0.311 | −0.308 | −0.478 | UN2 |
| ChestMNIST | 0.835 | 0.949 | 0.834 | 0.949 | −0.001 | 0.000 | CA |

**Table 7. Robustness on MedMNIST-C (lower is better; raw ratio scale).**

| Dataset | Clean score | BE | rBE |
|---|---|---|---|
| BloodMNIST | 0.012 | 2.103 | 2.654 |
| BreastMNIST | 0.204 | 1.591 | 3.243 |
| DermaMNIST | 0.376 | 1.296 | 3.020 |
| OCTMNIST | 0.202 | 1.256 | 1.715 |
| OrganAMNIST | 0.092 | 2.225 | 2.992 |
| OrganCMNIST | 0.141 | 3.352 | 6.289 |
| PathMNIST | 0.035 | 0.517 | 0.663 |
| PneumoniaMNIST | 0.093 | 2.085 | −6.210[†] |
| RetinaMNIST | 0.505 | 1.067 | 2.426 |
| TissueMNIST | 0.421 | 1.481 | 3.082 |
| OrganSMNIST | 0.272 | 1.256 | 2.065 |
| ChestMNIST | 0.461 | 1.424 | 2.930 |
| **Mean** | – | 1.640 | 2.070 |

[†] Negative rBE indicates BioFuse's error increases *less* than AlexNet's; here certain artefacts accentuate lung infiltrates, improving separability.

- Clean score – balanced error on the uncorrupted test set.

- BE – AlexNet-normalised balanced error averaged over all 55 corruption–severity pairs (AlexNet = 1).

- rBE – AlexNet-normalised *increase* in error relative to the clean set (AlexNet = 1). Values < 1 mean the model degrades less than AlexNet under corruption.

*Note:* the original paper multiplies these ratios by *100* (AlexNet = 100). We report the raw ratios from medmnistc-api for transparency; multiply by *100* to obtain the paper's scale.

**Discussion.** Averaged across all datasets, BioFuse achieves **BE = 1.64** and **rBE = 2.07**. This is higher (worse) than the single-backbone ViT-B/16 benchmark (BE ≈ 0.763) [80], mirroring the robustness–accuracy trade-off previously reported on CIFAR-C [81].

Fusion combines complementary features on clean data, but when corruptions affect the CNN and ViT backbones differently their representations can diverge, lowering performance. The clearest example is *OrganCMNIST* (BE = 3.35), where colour-contrast shifts likely drive conflicting cues between backbones. Conversely, on *PathMNIST* (BE < 0.6) all backbones respond similarly to texture-dominated corruptions, so fusion remains robust. Robustness is therefore highly dataset dependent.

Overall, our results underline a key design challenge: while fusion improves representational capacity on clean data, additional mechanisms are needed to maintain robustness. Future work will explore corruption-aware fusion, adaptive backbone weighting under distribution shift, and robustness-preserving training strategies for multi-backbone models.

## 5. Discussion

### 5.1 Key findings

Based on the comprehensive results in Table 3, we would like to highlight some important observations:

- **State-of-the-art performance.** BioFuse demonstrates exceptional performance on the MedMNIST+ benchmark, achieving new state-of-the-art (SOTA) scores across multiple datasets. Specifically, it achieves the highest AUC in five datasets and the best accuracy in two datasets. These improvements, achieved in a highly competitive benchmark, underscore BioFuse's ability to extract and combine more informative features from multiple pre-trained models.

- **Strong performance on DermaMNIST.** A notable improvement was observed on the DermaMNIST dataset, where BioFuse achieved state-of-the-art performance in both AUC and accuracy. Despite the SOTA AUC already being at 0.97, BioFuse managed to deliver a further performance improvement in AUC and ~1% in Accuracy. This demonstrates BioFuse's ability to improve performance even on tasks where the margin for improvement is small.

- **Near SOTA AUC performance.** BioFuse demonstrated highly competitive performance across multiple datasets, achieving AUC scores within 2% of state-of-the-art methods. This consistent performance was observed across various CT imaging tasks (OrganAMNIST, OrganCMNIST, OrganSMNIST) and microscopy datasets (TissueMNIST, PathMNIST). While BioFuse did not achieve the highest test accuracy for most datasets, it notably achieved SOTA performance on PathMNIST. The minimal gap in AUC scores compared to best-performing methods suggests BioFuse maintains robust class separation capabilities across these diverse medical imaging tasks.

- **Strong AUC scores even when accuracy suffers.** For datasets such as OCTMNIST, PneumoniaMNIST, and ChestMNIST, BioFuse's test accuracy lagged behind existing models, but it outperformed them in AUC scores. Specifically, BioFuse matched the SOTA AUC of 0.992 for OCTMNIST, 0.944 (vs. 0.939) for BreastMNIST, and 0.835 (vs. 0.822) for ChestMNIST, showing that even when accuracy lags, BioFuse's embeddings allow the model to make confident and accurate class distinctions.

- **Efficacy of model ensembles.** Across all datasets, the best-performing combinations involved multiple foundation models. No single model was able to outperform the ensembles, underscoring the importance of model diversity in feature representation. This finding suggests that the combined outputs of different models allow BioFuse to better capture complementary information across multiple modalities, resulting in more robust performance across a variety of biomedical imaging tasks.

### 5.2 Cross-Modal transfer

Biomedical foundation models exhibited remarkable cross-modal transfer capabilities, often excelling in tasks well beyond their original training domains. This generalisation likely stems from shared visual feature representations that transcend specific imaging modalities. Notably, models trained on a single modality (such as histopathology-specific models)

demonstrated stronger cross-modal transfer than those trained on multiple modalities, suggesting a potential benefit to focused, modality-specific pre-training for developing generalisable representations.

Histopathology models — particularly UNI, Hibou-B, and Prov-GigaPath — demonstrated exceptional cross-modal performance, likely due to their Vision Transformer-based architectures [38,40,41]. These models develop capacity to process multi-scale visual features during pre-training, from micron-scale cellular details to millimeter-scale tissue structures. This multi-scale capability appears particularly transferable across modalities, as the feature extraction mechanisms that identify cellular boundaries and tissue organization in histopathology may transfer effectively to detecting analogous structural patterns in retinal images and radiological scans, despite differences in visual appearance.

CheXagent's sophisticated architecture, which integrates an 8-billion parameter clinical language model with a vision encoder and vision-language bridge [46], may contribute to its strong cross-modal adaptability. Pre-trained on 6 million instruction-image-answer triplets across 65 diverse datasets, this model appears to develop both modality-specific and modality-agnostic feature representations. The language component potentially functions as a semantic intermediary, facilitating knowledge transfer between distinct imaging domains. This architectural advantage correlates with CheXagent's impressive performance in non-radiological tasks such as dermatology (AUC 0.960) and microscopy (AUC 0.999).

While our findings demonstrate substantial cross-modal capabilities, this transfer is not universal across all biomedical tasks. We observed performance decreases in highly specialized domains such as CT scan interpretation, where Organ-AMNIST and OrganCMNIST showed the widest performance gaps between specialized and cross-modal models. These limitations suggest that effective cross-modal transfer depends on both model architecture and the inherent similarity between source and target domains. Future work should explore these boundaries systematically, potentially guiding the development of more universally transferable biomedical foundation models.

## 5.3 Computational considerations

Understanding BioFuse's computational demands is essential for practical implementation. Following [83], we analyzed total response time as a realistic computational performance metric. Figure 6 decomposes the computational requirements for BioFuse experiments into three sequential stages: embedding extraction, exhaustive XGBoost evaluation across 511 combinations, and 100-trial Bayesian optimization of nine hyperparameters. Detailed per-dataset runtime tables is provided in S6 Table.

Hyperparameter optimization constitutes the primary computational requirement across all datasets. Large datasets like TissueMNIST (165K training samples) and those with many labels (ChestMNIST with 14 labels) require 70–80% of total runtime for optimization (approximately 35h and 40h, respectively) due to longer training times per trial, while the smallest datasets complete in under one hour. For small datasets like BreastMNIST and PneumoniaMNIST (several hundred images each), fixed-cost embedding extraction completes within minutes, making the 511-combination evaluation the second-largest computational component. Embedding extraction itself represents a bounded computational component: batched inference at 128 images per step maintains extraction below 7h even for the largest datasets when aggregated across all models. For individual models, extraction remains under 2h for most model-dataset combinations, with CheXagent on TissueMNIST (3.4h) and ChestMNIST (1.6h) representing the longest single-model extraction times (S7 Fig).

These results indicate that with cached embeddings, the primary computational bottleneck lies in downstream model selection. Beyond runtime considerations, we also examined GPU memory requirements to assess hardware accessibility. Our analysis showed that 7 of 9 foundation models operated within 5 GB of VRAM, making them suitable for consumer-grade GPUs. However, CheXagent, with 8 billion parameters, required 33.8 GB VRAM, necessitating server-grade GPU hardware (S8 Fig). Although memory requirements scale with dataset size, the relative differences between models remain consistent.

To support reproducibility and eliminate the need for time-intensive embedding extraction, we provide the complete embedding cache via Zenodo for MedMNIST+ [84] and ImageNet-1K [85].

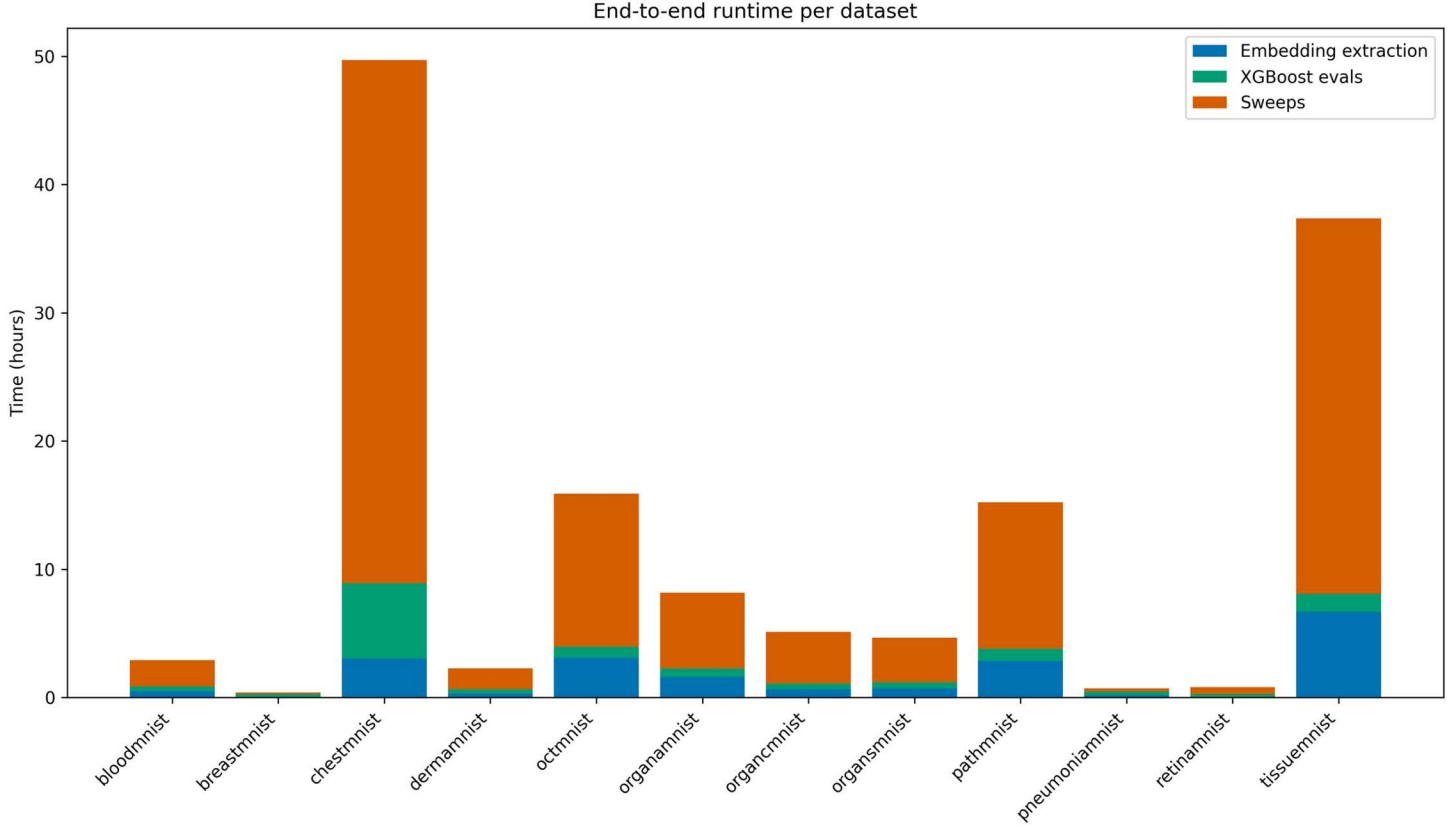

**Fig 6. End-to-end runtime per dataset.** Stacked bars show total wall-clock time (hours) to reproduce each BioFuse experiment. The blue segment indicates embedding extraction for the nine backbones, the green segment the XGBoost evaluation runs, and the orange segment the hyperparameter sweep on Weights & Biases.

### 5.4 Limitations and future directions

- **Evaluation scope and task generalisation.** BioFuse is currently evaluated only on classification tasks (binary, multi-class, and multi-label) using MedMNIST+. Testing on segmentation, detection, regression, and real-world clinical data would give a fuller picture. Rigorous out-of-distribution evaluation on clean external datasets remains important future work. Future work could also extend BioFuse to time-series modalities and multimodal clinical records.

- **Dependency on pre-trained models.** Performance is bounded by the quality and diversity of the underlying backbones. Expanding the pool to MRI, ultrasound, genomics and clinical-text models would reduce bias and enable broader multimodal fusion. Systematic bias evaluation across demographic groups remains essential future work.

- **Computational overhead.** Extracting embeddings from many large backbones (e.g., CheXagent, 33 GB VRAM) and running exhaustive searches is GPU-intensive. More efficient search heuristics or progressive fusion strategies are an important next step.

- **Interpretability challenges.** Combining multiple backbones obscures how each modality contributes to a prediction. Integrating saliency maps, feature attribution or attention visualisation will be critical for clinical trust and regulatory acceptance.

By addressing these areas through collaboration between AI researchers and clinicians, BioFuse can become a more efficient, interpretable and generalisable tool for multimodal biomedical applications.

## 6. Conclusion

The integration of diverse foundation models represents a promising frontier for advancing biomedical imaging analysis. In this work, we introduced BioFuse, a novel framework that systematically fuses embeddings from multiple biomedical foundation models to generate optimised representations for downstream tasks. Evaluated across 12 diverse imaging modalities in the MedMNIST+ benchmark, BioFuse with XGBoost classification outperformed existing methods, achieving the highest AUC in five datasets and maintaining near-SOTA performance in most others. Notably, it demonstrated exceptional performance in dermatology classification (DermaMNIST) and revealed unexpected cross-modal transfer capabilities in histopathology and radiology models like UNI, Hibou-B, Prov-GigaPath, and CheXagent.

These results highlight the benefits of leveraging multiple pre-trained models rather than relying on a single foundation model. BioFuse's ability to automatically identify and integrate complementary representations from diverse models suggests significant potential for healthcare applications requiring comprehensive image interpretation across modalities. The framework's extensible architecture ensures adaptability to future foundation models as they emerge.

While demonstrating clear advantages, BioFuse faces challenges, including computational overhead from grid search and potential redundancy in concatenated embeddings. Future work should explore more efficient fusion strategies, expand applications beyond classification to segmentation and detection tasks, and incorporate interpretability mechanisms essential for clinical adoption and regulatory approval.

By harnessing the collective strengths of multiple foundation models through a systematic approach to embedding fusion, BioFuse not only improves performance on benchmark tasks but also opens new avenues for cross-modal knowledge transfer in biomedical imaging. This contribution moves us closer to more reliable and comprehensive AI-assisted medical decision-making systems that can integrate information across the diverse imaging modalities encountered in clinical practice.

## Supporting information

**S1 Appendix. BioFuse API workflow.**
(PDF)

**S2 Table. XGBoost hyperparameter search ranges used for Bayesian optimization.**
(PDF)

**S3 Table. Optimal XGBoost hyperparameter configurations for concatenation fusion for all MedMNIST+ datasets.**
(PDF)

**S4 Table. Optimal XGBoost hyperparameter configurations for self-attention fusion for all MedMNIST+ datasets.**
(PDF)

**S5 Table. Optimal XGBoost hyperparameter configurations for oracle-single models using best individual backbones for all MedMNIST+ datasets.**
(PDF)

**S6 Table. Computational runtime breakdown for the concatenation fusion experiment, showing embedding extraction, model training, hyperparameter sweep, and total processing time for each dataset.**
(PDF)

**S7 Fig. Heatmap of embedding extraction times (hours) across nine foundation models and twelve MedMNIST+ datasets.**
(PDF)

**S8 Fig. Peak GPU VRAM usage (GB) for each foundation model during embedding extraction.**
(PDF)

**S9 Appendix. Pretraining dataset details for all nine biomedical foundation models used in BioFuse.**
(PDF)

**S10 Appendix. MedMNIST+ dataset descriptions including imaging modalities, clinical tasks, and dataset statistics.**
(PDF)

## Author contributions

**Conceptualization:** Mirza Nasir Hossain.

**Data curation:** Mirza Nasir Hossain.

**Formal analysis:** Mirza Nasir Hossain.

**Investigation:** Mirza Nasir Hossain.

**Methodology:** Mirza Nasir Hossain.

**Project administration:** David Harris-Birtill.

**Software:** Mirza Nasir Hossain.

**Supervision:** David Harris-Birtill.

**Validation:** Mirza Nasir Hossain.

**Visualization:** Mirza Nasir Hossain.

**Writing – original draft:** Mirza Nasir Hossain.

**Writing – review & editing:** David Harris-Birtill.

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
