## [Decision Letter · Decision Letter 0]

22 May 2025

Dear Dr. Hossain,

Thank you for submitting your manuscript to PLOS ONE. After careful consideration, we feel that it has merit but does not fully meet PLOS ONE’s publication criteria as it currently stands. Therefore, we invite you to submit a revised version of the manuscript that addresses the points raised during the review process.

We look forward to receiving your revised manuscript.

Kind regards,

Xu Yanwu

Academic Editor

PLOS ONE

**Journal Requirements:**

1. When submitting your revision, we need you to address these additional requirements. Please ensure that your manuscript meets PLOS ONE's style requirements, including those for file naming. The PLOS ONE style templates can be found at https://journals.plos.org/plosone/s/file?id=wjVg/PLOSOne_formatting_sample_main_body.pdf and https://journals.plos.org/plosone/s/file?id=ba62/PLOSOne_formatting_sample_title_authors_affiliations.pdf 2. Thank you for stating the following in the Acknowledgments Section of your manuscript: We thank the School of Computer Science at the University of St Andrews for providing computational resources and support. We also appreciate the IT team for their technical assistance. We note that you have provided funding information that is not currently declared in your Funding Statement. However, funding information should not appear in the Acknowledgments section or other areas of your manuscript. We will only publish funding information present in the Funding Statement section of the online submission form. Please remove any funding-related text from the manuscript and let us know how you would like to update your Funding Statement. Currently, your Funding Statement reads as follows: The author(s) received no specific funding for this work. Please include your amended statements within your cover letter; we will change the online submission form on your behalf. 3. Please note that your Data Availability Statement is currently missing the repository name. If your manuscript is accepted for publication, you will be asked to provide these details on a very short timeline. We therefore suggest that you provide this information now, though we will not hold up the peer review process if you are unable. 4. Your ethics statement should only appear in the Methods section of your manuscript. If your ethics statement is written in any section besides the Methods, please move it to the Methods section and delete it from any other section. Please ensure that your ethics statement is included in your manuscript, as the ethics statement entered into the online submission form will not be published alongside your manuscript. 5. We notice that your supplementary Appendix’s are included in the manuscript file. Please remove them and upload them with the file type 'Supporting Information'. Please ensure that each Supporting Information file has a legend listed in the manuscript after the references list.

**Additional Editor Comments:**

Please follow reviewers' comments to improve your manuscript

Reviewers' comments:

Reviewer's Responses to Questions

**Comments to the Author**

1. Is the manuscript technically sound, and do the data support the conclusions?

Reviewer #1: Yes

Reviewer #2: Yes

2. Has the statistical analysis been performed appropriately and rigorously?

Reviewer #1: Yes

Reviewer #2: Yes

3. Have the authors made all data underlying the findings in their manuscript fully available?

Reviewer #1: Yes

Reviewer #2: Yes

4. Is the manuscript presented in an intelligible fashion and written in standard English?

Reviewer #1: Yes

Reviewer #2: Yes

**Reviewer #1:** This manuscript proposed BioFuse, a framework for efficient multimodal fusion in biomedical image classification, leveraging vector concatenation and automated model selection via XGBoost. BioFuse concatenates embeddings from multiple pretrained models (e.g., BioMedCLIP, UNI) along the feature dimension, preserving modality-specific information without introducing additional parameters. In the medmnist+ benchmark test, the number of biofuse in five Numbers was set up to the highest AUC, which was 3.2% on average, and the cross-modal fusion significantly enhanced performance.

In general, this manuscript is prepared with good novelty and solid data. However, prior to publication, the authors are encouraged to consider the following comments.

(1)The manuscript only assessed the performance of BioFuse through AUC and accuracy, and does not involve other key indicators (such as computational efficiency, model robustness, and cross-dataset generalization ability). For example, the robustness of the model to noisy data or abnormal samples was not discussed. It is suggested to supplement multi-dimensional performance comparisons to comprehensively verify the practicability of the method.

(2)It is suggested to supplement and elaborate in the manuscript the possible feature conflict problems in multimodal feature fusion (such as the scale differences embedded in different models), and how to handle such problems.

(3)The manuscript mentions the selection of model parameters, but does not explicitly list specific parameter values (such as the learning rate of XGBoost, maximum depth, etc.) and their impact on the results. It is suggested to provide a complete hyperparameter configuration table and discuss the parameter sensitivity analysis.

(4)It is suggested that ablation experiments should be added to the manuscript to verify the independent contribution of key designs in BioFuse (such as vector join strategies and automatic model selection) to performance, quantify the improvement ratio of each part to the final result, and compare it with other fusion methods.

**Reviewer #2:**  The manuscript introduces BioFuse, a novel framework for fusing embeddings from multiple biomedical foundation models, demonstrating improved performance on the MedMNIST+ benchmark. To enhance the study, consider expanding the evaluation to include segmentation, object detection, and regression tasks, as well as testing on real-world clinical datasets. Additionally, incorporating a more diverse set of foundation models from various biomedical domains and exploring alternative fusion strategies, such as attention-based methods, could further improve the framework's versatility and efficiency. Addressing computational efficiency and enhancing interpretability through explainability techniques would also be valuable. Finally, ensuring the models are representative and unbiased, along with providing comprehensive documentation and tutorials, would make BioFuse more accessible and impactful for a broader audience.

**Do you want your identity to be public for this peer review?** For information about this choice, including consent withdrawal, please see our Privacy Policy

Reviewer #1: No

Reviewer #2: No

---

## [Author Response · Author response to Decision Letter 1]

4 Aug 2025

Editor comments:

Journal requirements addressed:

1. Style requirements: Manuscript now follows PLOS ONE formatting guidelines.

2. Funding statement: We have removed the acknowledgment of computational resources from the manuscript as requested. We would like to update our Funding Statement to accurately reflect that this work was supported by a PhD studentship. Please update the online submission form to:

"This study was supported by a PhD studentship from the School of Computer Science, University of St Andrews. The funder had no role in study design, data collection and analysis, decision to publish, or preparation of the manuscript."

3. Data availability: Repository details have been added to the Data Availability Statement.

4. Ethics statement: Now appears only in the Methods section.

5. Supporting information: All supplementary materials have been removed from the main manuscript and uploaded as separate Supporting Information files with appropriate legends.

Reviewer comments:

We thank the reviewers for their constructive and insightful comments, which have substantially improved our manuscript. We have addressed all concerns as detailed below. Major revisions include: (1) comprehensive robustness evaluation using the MedMNIST-C corruption benchmark, (2) detailed computational efficiency analysis with runtime breakdowns, (3) two new ablation studies comparing fusion strategies, and (4) a new modular code repo with enhanced documentation. All line numbers refer to the new manuscript. Our replies in blue.

Response to Reviewer #1

(1) The manuscript only assessed the performance of BioFuse through AUC and accuracy, and does not involve other key indicators (such as computational efficiency, model robustness, and cross-dataset generalization ability). For example, the robustness of the model to noisy data or abnormal samples was not discussed. It is suggested to supplement multi-dimensional performance comparisons to comprehensively verify the practicability of the method.

1. Robustness evaluation: We have added a comprehensive robustness analysis using the MedMNIST-C benchmark, which applies 11 medically-motivated corruptions at 5 severity levels to all MedMNIST+ datasets. We report balanced error (BE) and relative balanced error (rBE) following the MedMNIST-C evaluation protocol (new section "Robustness to realistic image corruptions", lines 532, Table 7).

2. Computational Efficiency: We have substantially expanded our computational analysis. Due to a pipeline error discovered during revision (where 28×28 images were upscaled to 224×224 instead of using the native 224×224 MedMNIST+ images), we re-executed all experiments with corrected preprocessing. This provided an opportunity to collect detailed timing data across three phases: embedding extraction, model evaluation, and hyperparameter optimization. Results are presented in:

• Figure 6: End-to-end runtime visualization for all datasets

• S6 Table: Detailed per-dataset computational breakdown

• S7 Fig: Heatmap of embedding extraction times for each model-dataset pair

3. Cross-dataset generalisation: We investigated several public datasets (BUSI, ISIC, RSNA) for out-of-distribution evaluation but discovered that most are included in the pre-training corpora of our foundation models (detailed in S9 Appendix), rendering them unsuitable as independent test sets. We acknowledge this limitation and identify rigorous out-of-distribution evaluation as an important direction for future work (Limitations section, lines 693-).

(2) It is suggested to supplement and elaborate in the manuscript the possible feature conflict problems in multimodal feature fusion (such as the scale differences embedded in different models), and how to handle such problems.

We have added a new subsection "Feature-scale alignment" (lines 267) within the Fusion methodology section. This addition explains that: (i) vector concatenation preserves each model's features in isolated sub-blocks, avoiding cross-model normalization requirements, and (ii) our downstream classifier (XGBoost) makes splitting decisions based primarily on relative feature ordering rather than absolute magnitudes, providing natural robustness to scale differences. We cite relevant literature [56,57,58] supporting this scale-invariance property of tree-based methods.

(3) The manuscript mentions the selection of model parameters, but does not explicitly list specific parameter values (such as the learning rate of XGBoost, maximum depth, etc.) and their impact on the results. It is suggested to provide a complete hyperparameter configuration table and discuss the parameter sensitivity analysis.

We now provide complete hyperparameter configurations in:

• S2 Table: XGBoost hyperparameter search ranges used for Bayesian optimization

• S3 Table: Optimal parameters for concatenation fusion (main results)

• S4 Table: Optimal parameters for self-attention fusion ablation

• S5 Table: Optimal parameters for oracle-single ablation

Each table lists all nine XGBoost hyperparameters for each of the 12 datasets. Regarding sensitivity analysis, we note that exhaustive parameter sensitivity sweeps across nine hyperparameters, 12 datasets, and three fusion variants would require thousands of additional training cycles, exceeding our computational resources. Consistent with established MedMNIST+ benchmarking practices, we report optimal configurations discovered through Bayesian optimization.

(4) It is suggested that ablation experiments should be added to the manuscript to verify the independent contribution of key designs in BioFuse (such as vector join strategies and automatic model selection) to performance, quantify the improvement ratio of each part to the final result, and compare it with other fusion methods.

We have added two comprehensive ablation studies:

1. Self-attention fusion vs. Concatenation (Table 5, lines 482-): This ablation replaces simple concatenation with a learnable multi-head self-attention mechanism.

2. Oracle-single vs. Concatenation (Table 6, lines 517-): This ablation compares fusion against using only the single best model per dataset.

Both ablations include detailed discussions of the findings and their implications for the fusion strategy.

Response to Reviewer #2

(1) To enhance the study, consider expanding the evaluation to include segmentation, object detection, and regression tasks, as well as testing on real-world clinical datasets.

We appreciate this suggestion for broadening BioFuse's evaluation. As noted in our response to Reviewer #1 (Comment 1.3), we investigated several clinical datasets but found that most are included in our foundation models' pre-training corpora (S9 Appendix) or serve as sources for MedMNIST+ (S10 Appendix). Regarding task diversity, we acknowledge that extending to segmentation and object detection represents an important future direction. Our Limitations section (lines 692-) to explicitly identify these as priorities for future work, while noting that such extensions would require substantial architectural modifications beyond the current scope.

(2) Additionally, incorporating a more diverse set of foundation models from various biomedical domains and exploring alternative fusion strategies, such as attention-based methods, could further improve the framework's versatility and efficiency.

We have addressed both aspects of this comment:

1. Alternative Fusion Strategy: We implemented and evaluated a self-attention fusion mechanism as requested (Table 5, lines 482-). This learnable approach underperformed simple concatenation on most datasets, providing empirical validation for our design choice.

2. Model Diversity: Our current pool of nine foundation models already spans diverse biomedical domains (histopathology, radiology, multimodal vision-language) and architectures. We discuss future incorporation of MRI, ultrasound, genomics, and clinical text models in the Limitations section (lines 699-). The framework's modular architecture facilitates easy integration of new models as they become available.

(3) Addressing computational efficiency and enhancing interpretability through explainability techniques would also be valuable.

Computational Efficiency: Please see our detailed response to Reviewer #1 (Comment 1.2). We have added comprehensive runtime analysis including wall-clock measurements across all computational phases (Figure 6, S6 Table, S7 Figure) and GPU memory requirements (S8 Figure). We discuss this in depth in the "Computational considerations" section (lines 654-).

Interpretability: We acknowledge interpretability as a critical requirement for clinical deployment and discuss this in the limiations section (lines 706-) and identify the integration of saliency maps, feature attribution methods, and attention visualization as essential future work for clinical acceptance and regulatory approval.

(4) Finally, ensuring the models are representative and unbiased, along with providing comprehensive documentation and tutorials, would make BioFuse more accessible and impactful for a broader audience.

We have taken several steps to improve BioFuse's accessibility:

1. Enhanced Architecture & Documentation: We are actively developing improved documentation and code architecture. A new modular version of the codebase (https://github.com/mnhcorp/biofuse/tree/v1.0) is currently in progress and will include:

• A modular pluggable architecture

• Detailed architecture documentation

• Installation guides

• API reference documentation

• Example notebooks demonstrating common use cases

While this enhanced version is still under development, the current release provides functional code for reproducing all experiments. Additionally, we have made pre-computed embedding caches available on Zenodo to eliminate computational barriers for users.

2. Bias and Representativeness: We acknowledge the importance of model bias assessment. While our nine foundation models were trained on diverse, large-scale datasets (S9 Appendix), systematic bias evaluation across demographic groups remains future work. We have added this consideration to our Limitations section (lines 700-).

---

## [Decision Letter · Decision Letter 1]

11 Dec 2025

Dear Dr. Hossain,

Thank you for submitting your manuscript to PLOS ONE. After careful consideration, we feel that it has merit but does not fully meet PLOS ONE’s publication criteria as it currently stands. Therefore, we invite you to submit a revised version of the manuscript that addresses the points raised during the review process.

We look forward to receiving your revised manuscript.

Kind regards,

Xu Yanwu

Academic Editor

PLOS One

Journal Requirements:

Additional Editor Comments:

The hierarchical structure of the manuscript appears somewhat disorganized. To enhance readability and facilitate readers' comprehension of the logical flow, it is recommended to add a sequential numbering system to the subheadings (e.g., 1, 1.1, 1.1.1). Additionally, certain sections could be merged for better coherence。 The "Background" and "Related Works" sections may be integrated into a single section. Similarly, the "Bisfuse" section is advised to be merged into the "Methods" section.

Reviewers' comments:

Reviewer's Responses to Questions

**Comments to the Author**

Reviewer #1: (No Response)

2. Is the manuscript technically sound, and do the data support the conclusions?

Reviewer #1: (No Response)

3. Has the statistical analysis been performed appropriately and rigorously?

Reviewer #1: (No Response)

4. Have the authors made all data underlying the findings in their manuscript fully available?

Reviewer #1: (No Response)

5. Is the manuscript presented in an intelligible fashion and written in standard English?

Reviewer #1: (No Response)

Reviewer #1: In my opinion, the authors have answered all the questions from the last round and the paper can be accepted in the current version.

**Do you want your identity to be public for this peer review?** For information about this choice, including consent withdrawal, please see our Privacy Policy

Reviewer #1: No

---

## [Author Response · Author response to Decision Letter 2]

21 Jan 2026

Response to Academic Editor comments

1. Sequential numbering: Implemented a consistent numbering scheme across headings/subheadings (e.g., 1, 1.1, 1.1.1) to improve readability and navigation.

2. Merge Background + Related work: Implemented by consolidating into Section 2: Background and related work, with background context in Section 2.1 and related work in Section 2.2 for a clearer narrative flow.

3. Merge BioFuse into Methods: Implemented by integrating the BioFuse description into Methods as Section 3.2: BioFuse framework, alongside ethics, datasets, and experiments.

4. Experimental section structure: Consolidated the former “Experimental design” and “Experimental procedure” into a single Experiments section with clearer internal hierarchy; the Objectives subsection now explicitly enumerates robustness and ablation analyses consistent with Results.

5. Cross-references: Updated internal references throughout to match the revised section structure.

(An expanded point-by-point response is also provided in the uploaded “Response to Reviewers” document.)

---

## [Decision Letter · Decision Letter 2]

16 Feb 2026

BioFuse: an embedding fusion framework for biomedical foundation models

PONE-D-25-10720R2

Dear Dr. Hossain,

We’re pleased to inform you that your manuscript has been judged scientifically suitable for publication and will be formally accepted for publication once it meets all outstanding technical requirements.

Kind regards,

Xu Yanwu

Academic Editor

PLOS One

Additional Editor Comments (optional):

Reviewers' comments:

Reviewer's Responses to Questions

**Comments to the Author**

Reviewer #3: All comments have been addressed

2. Is the manuscript technically sound, and do the data support the conclusions?

Reviewer #3: Yes

3. Has the statistical analysis been performed appropriately and rigorously?

Reviewer #3: N/A

4. Have the authors made all data underlying the findings in their manuscript fully available?

Reviewer #3: No

5. Is the manuscript presented in an intelligible fashion and written in standard English?

Reviewer #3: Yes

Reviewer #3: The revised version of the manuscript is now in a well-organized pattern and suitable for publication.

**Do you want your identity to be public for this peer review?** For information about this choice, including consent withdrawal, please see our Privacy Policy

Reviewer #3: No

---

## [Editor Report · Acceptance letter]

PONE-D-25-10720R2

PLOS One

Dear Dr. Hossain,

I'm pleased to inform you that your manuscript has been deemed suitable for publication in PLOS One. Congratulations! Your manuscript is now being handed over to our production team.

Kind regards,

on behalf of

Dr. Xu Yanwu

Academic Editor

PLOS One